# Structural mechanisms of assembly, gating, and calmodulin modulation of human olfactory CNG channel

Jing Xue [1,2,3] ✉, Ninghai Gan [1,2,3], Weizhong Zeng[1,2] & Youxing Jiang [1,2] ✉

Mammalian cyclic nucleotide-gated (CNG) channels play crucial roles in visual and olfactory signal transduction. In olfactory sensory neurons, the native CNG channel functions as a heterotetramer consisting of CNGA2, CNGA4, and CNGB1b subunits and is activated by cAMP. Calmodulin (CaM) modulates the activity of the olfactory CNG channel, enabling rapid adaptation to odorants. Here we present cryo-EM structures of the native human olfactory CNGA2/A4/B1b channel in both CaM-bound closed and cAMP-bound open states, elucidating the molecular basis of the 2:1:1 subunit stoichiometry in channel assembly and the asymmetrical channel gating upon cAMP activation. Combining structural and functional analyses with AlphaFold prediction, we define two distinct CaM binding sites (CaM1 and CaM2) on the N- and C-terminal regions of CNGB1b, respectively, shedding light on the molecular mechanism of $Ca^{2+}$/CaM-mediated rapid inhibition of the native olfactory CNG channel.

Mammalian cyclic nucleotide-gated (CNG) channels play central roles in signal transduction in photoreceptor and olfactory sensory neurons[1–3]. Belonging to the voltage-gated ion channel (VGIC) superfamily, CNG channels are non-selective tetrameric cation channels that are activated by cyclic nucleotides rather than voltage[4]. The mammalian CNG channel family consists of four CNGA subunits (CNGA1-CNGA4) and two CNGB subunits (CNGB1 and CNGB3)[3,5]. The three principal components of CNG channels, namely rod CNGA1, olfactory CNGA2, and cone CNGA3, can form functional homotetrameric channels in heterologous expression systems. However, native CNG channels function as heterotetrameric complexes consisting of CNGA and CNGB subunits[2,6–12].

The native CNG channels in rod or cone photoreceptors are heterotetramers of CNGA1/B1 or CNGA3/B3, respectively, with a subunit stoichiometry of 3:1[13–17]. In contrast, the native CNG channel in olfactory sensory neurons has a subunit composition of two CNGA2, one CNGA4, and one CNGB1b, an olfactory-specific shorter splice variant of CNGB1 lacking the N-terminal glutamic acid-rich protein domain (GARP)[18–20]. Under physiological conditions, photoreceptor CNG channels are activated by intracellular cGMP, whereas olfactory CNG

channels are activated by cAMP[3]. Upon activation, CNG channels conduct inward $Na^+$ and $Ca^{2+}$ currents, thereby depolarizing the membrane potential and increasing the intracellular $[Ca^{2+}]$. The elevated intracellular $Ca^{2+}$ levels can, in turn, trigger the $Ca^{2+}$/calmodulin (CaM)-dependent desensitization of CNG channels, a negative feedback process critical for light adaptation in photoreceptors and odor adaptation in olfactory sensory neurons[21–23].

Although the olfactory CNGA2 subunit can form a functional homomeric channel, the presence of CNGA4 and CNGB1b subunits in the native olfactory CNG render the channel with several distinct features in cAMP activation and $Ca^{2+}$/CaM-induced desensitization: the native olfactory CNG exhibits much higher sensitivity to cAMP than the CNGA2 homomer; CaM is tightly associate with the native olfactory CNG even in the absence of $Ca^{2+}$; $Ca^{2+}$/CaM induces faster inhibition in the native channel; the native channel is more sensitive to $Ca^{2+}$/CaM inhibition, resulting in more than 10-fold decrease in its apparent cAMP affinity[18–21,24–27]. Multiple potential CaM-binding sites have been suggested across the three subunits of olfactory CNG, including the N-terminal region of A2, and both N- and C-terminal regions of B1b. However, only the N-terminal site on B1b has been functionally

[1]Howard Hughes Medical Institute and Department of Physiology, University of Texas Southwestern Medical Center, Dallas, TX, USA. [2]Department of Biophysics, University of Texas Southwestern Medical Center, Dallas, TX, USA. [3]These authors contributed equally: Jing Xue, Ninghai Gan.
✉e-mail: jxue@shsmu.edu.cn; youxing.jiang@utsouthwestern.edu

demonstrated to be indispensable for CaM binding and inhibition in the native olfactory channel[24,28–31].

While the olfactory CNG channel has been extensively studied for many years with a plethora of biochemical and physiological data, its structural information is lagging behind the other members of CNG channels in photoreceptors[16,17,32,33]. The molecular mechanisms underlying the unique 2:1:1 subunit assembly, cAMP activation, and $Ca^{2+}$/CaM inhibition of the native olfactory CNG channel remain elusive. In this study, we purify the human olfactory CNG as a CNGA2/A4/B1b heterotetrameric complex and determine its structures in both the CaM-bound closed state and the cAMP-bound open state, revealing the structural basis of subunit assembly and cAMP activation in the native olfactory CNG. Furthermore, by combining structural and functional analyses with AlphaFold prediction, we define the CaM-binding sites in the CNGA2/A4/B1b assembly, shedding light on the molecular mechanism of $Ca^{2+}$/CaM-mediated rapid inhibition of the native olfactory CNG channel.

## Results

### Electrophysiology of human olfactory CNGA2/A4/B1b heterotetramer

To investigate the structure and function of the human olfactory CNG channel in native assembly, we generated an expression construct containing the genes of CNGA2, CNGA4, and CNGB1b (a shorter CNGB1 variant containing amino acids 456-1251), in a pEZT-BM expression vector (Method). In this construct, two neighboring genes were linked by a P2A 'self-cleaving' peptide that induces ribosomal skipping during translation[34]. The expression of this tri-cistronic construct in HEK293F cells using the BacMam system yielded a CNGA2/A4/B1b complex in 2:1:1 stoichiometry. The same construct in the pEZT-BM vector was also transfected into HEK293 cells for functional characterization using patch-clamp recordings in both whole-cell and inside-out configurations. The expressed CNGA2/A4/B1b channel exhibits similar biophysical properties to the native olfactory CNG - it can be activated by cytosolic cAMP with an $EC_{50}$ of ~5 μM and blocked by extracellular $Ca^{2+}$ with a $K_i$ of about 160 μM (Fig. 1a,b)[21,27,35]. In addition, the CNGA2/A4/B1b complex also exhibits the same high-affinity CaM binding and $Ca^{2+}$/CaM-mediated inhibition properties as the native olfactory CNG (Fig. 1c-e)[24]. As shown in Fig. 1c, adding $Ca^{2+}$ in the bath solution (cytosolic) induces rapid inhibition of CNGA2/A4/B1b conduction mediated by the endogenous CaM. Removing the bound CaM requires a long washout (~10 min) with EGTA-containing bath solution (Supplementary Fig. 1a), indicating its high-affinity binding to the channel complex. After washout, the channel can regain its inhibition by supplementing purified CaM protein with $Ca^{2+}$, confirming that the channel inhibition is mediated by $Ca^{2+}$/CaM (Fig. 1d). Similar to the endogenous CaM, the supplemented CaM has high-affinity binding to the channel and its removal requires a long washout with EGTA-containing solution (Supplementary Fig. 1b). Thus the expressed CNGA2/A4/B1b complex recapitulates the key functional features of native olfactory CNG and its structure represents the native channel assembly.

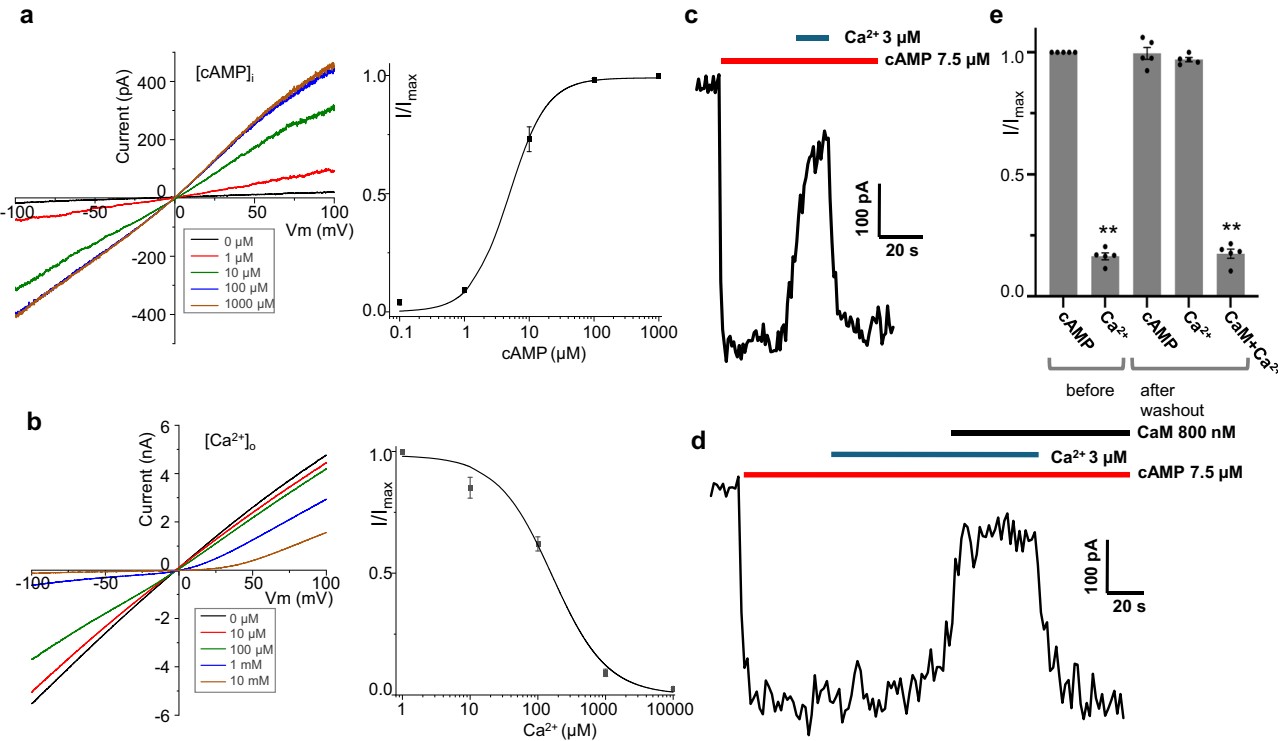

**Fig. 1 | Functional characterization of human CNGA2/A4/B1b. a** Sample I-V curves (left panel) recorded in excised patches with varying cAMP concentrations in the bath (cytosolic). Currents at 100 mV were used to generate the concentration-dependent cAMP activation curve (right panel). Curve is a least square fit to the Hill equation with $EC50 = 4.9 \pm 0.5$ μM, and $n = 1.51 \pm 0.15$. Data points are mean ± SEM ($n = 5$ independent replicates). **b** Sample I-V curves (left panel) recorded using patch clamp in the whole-cell configuration with varying $Ca^{2+}$ concentrations in the bath (extracellular). The pipette solution contains 1 mM cAMP. Currents at −100 mV were used to generate the concentration-dependent $Ca^{2+}$ inhibition curve (right panel). Curve is a least square fit to the Hill equation with

$K_i = 161.7 \pm 40.9$ μM and $n = 1.56 \pm 0.36$. Data points are mean ± SEM ($n = 5$ independent replicates). **c** Sample trace of endogenous CaM-mediated cytosolic $Ca^{2+}$ inhibition of CNGA2/A4/B1b recorded in excised patch. Inward current at −100 mV was elicited by 7.5 μM cAMP. **d** Sample trace of $Ca^{2+}$/CaM inhibition of CNGA2/A4/B1b recorded using the same patch as **(c)** after 10 min washout with EGTA to remove the endogenous CaM. **e** $Ca^{2+}$ and $Ca^{2+}$/CaM inhibition of CNGA2/A4/B1b before and after washout. Currents are normalized against the initial cAMP-activated inward currents at −100 mV. Data points are mean ± SEM ($n = 5$ independent replicates, $p$-values were calculated using a two-sided Student's t-test, ** $p < 0.01$). Source data are provided as a Source Data file.

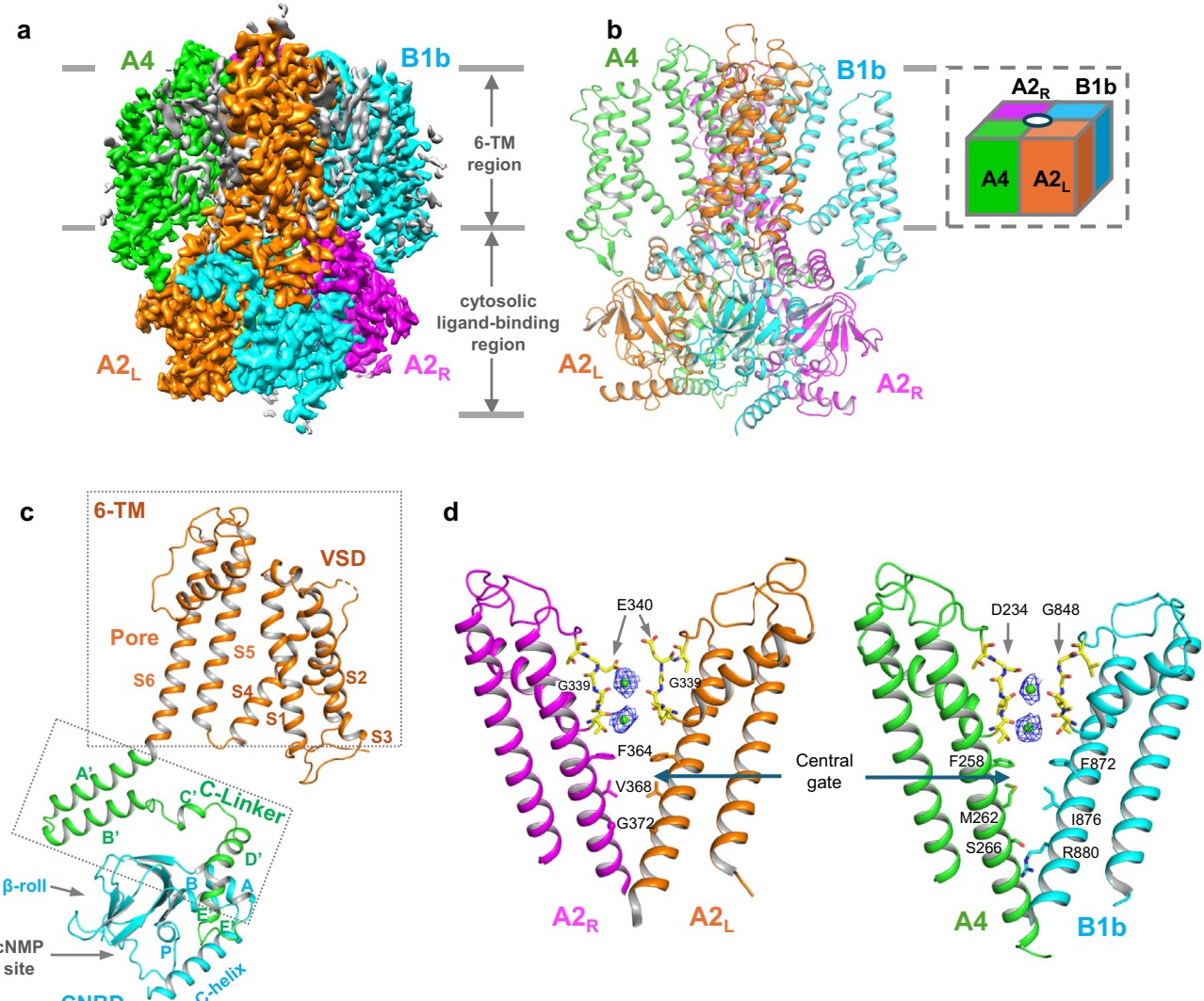

**Fig. 2 | Overall structure of human CNGA2/A4/B1b. a** Side view of 3D reconstruction of the core part of the CaM-bound closed CNGA2/A4/B1b with each subunit individually colored. Lipid density is shown in grey. The poorly defined C-terminal CLZ region and the bound CaM are not shown here. **b** Side view of cartoon representation of CNGA2/A4/B1b structure with subunit arrangement shown in the inset. **c** Structure of the A2L subunit with each domain individually colored. All other subunits have the identical domain arrangement. **d** Side views of the closed ion-conduction pores with two diagonal subunits illustrating the asymmetrical ion conduction pathway of CNGA2/A4/B1b. The density (blue mesh) of the filter Ca²⁺ (green spheres) is contoured at 8σ. Key filter and gating residues are shown in stick.

## Structural determination of human olfactory CNGA2/A4/B1b in complex with CaM

To minimize the heterogeneity in channel complex formation caused by the binding of endogenous CaM to the overexpressed CNGA2/A4/B1b, CaM was co-expressed with the CNGA2/A4/B1b construct and a homogenous CNGA2/A4/B1b-CaM complex was purified in the presence of $Ca^{2+}$ for structure determination (Method). The single particle cryo-EM structure of CNGA2/A4/B1b-CaM complex in an apo (in the absence of cAMP ligand), closed conformation was determined to a resolution of 2.83 Å (Fig. 2, Supplementary Figs. 2–4, and Supplementary Table 1). The high-quality EM density maps allowed accurate model building for major parts of CNG subunits, including residues 130-581 for the A2 subunit, residues 25-476 for the A4 subunit, and residues 645-1084 for the B1b subunit. The well-structured region from each subunit constitutes the core part of the CNG channel, containing the 6-TM transmembrane domain, the C-linker domain, and the C-terminal cytoplasmic cyclic nucleotide-binding domain (CNBD) (Fig. 2a–c). The C-terminal post-CNBD regions of all olfactory CNG subunits where CaM binds are poorly defined due to their flexibility,

and a focus refinement at these regions was performed to obtain structural information as discussed later.

In the CNGA2/A4/B1b heterotetramer, the two principal A2 sub-units are positioned diagonally rather than adjacently, yielding an A2-B1b-A2-A4 complex assembly (Fig. 2a,b). Consequently, the two A2 subunits pair with their respective neighboring subunits oppositely: one has A4 and B1 on the left and right sides, respectively, and the other way around for the other A2. As described in the next section, the two A2 subunits also adopt slightly different structures in the filter region and undergo different conformational changes upon channel opening. We therefore follow the same labeling method used for rod CNGA1/B1 heterotetramer and designate the two A2 subunits as A2L and A2R, respectively, based on their relative position to the B1b subunit (Left and Right) (Fig. 2b)[17]. In the closed CNGA2/A4/B1b, the core part of each subunit shares a similar overall structure with an identical domain arrangement (Fig. 2c and Supplementary Fig. 5). This high degree of structural homology among the A2, A4, and B1b subunits is likely essential for the stable assembly of the heterotetrameric channel.

## Ion conduction pore of CNGA2/A4/B1b

Despite similar overall structures among all subunits, differences in several key pore-lining residues yield an asymmetrical ion conduction pore in CNGA2/A4/B1b (Fig. 2d). In the selectivity filter, A2 and A4 contain an acidic residue (Glu340 in A2 and Asp234 in A4) at the external entrance of the channel important for $Ca^{2+}$ binding within the filter that blocks monovalent currents[32,35,36]. The equivalent residue becomes Gly in the B1b subunit. The absence of the acidic residue in B1b or B1 contributes to a reduced $Ca^{2+}$ blocking affinity in the heteromeric CNGA2/A4/B1b or CNGA1/B1 channels as compared to the homomeric CNGA2 or CNGA1 channels[12,17,35]. The Glu340 side chains of the two A2 subunits also orient differently in CNGA2/A4/B1b, with one pointing toward the central axis and the other pointing upward toward the external solution. Additionally, the filters of the two A2 subunits adopt a different main-chain configuration at Gly339 (Fig. 2d). With the structure determined in the presence of $Ca^{2+}$, the densities of two bound $Ca^{2+}$ ions are resolved inside the filter. Because of the asymmetrical filter structure, both ions are positioned off-center from the central axis (Fig. 2d).

Along the ion conduction pathway, two layers of hydrophobic residues on S6 helices form the central gate of the olfactory CNG in the middle of the membrane (Fig. 2d). One layer consists of four phenylalanine residues from all subunits (Phe364 from A2, Phe258 from A4, and Phe872 from B1b), whereas the other layer consists of different residues, including Val368 from A2, Met262 from A4, and Ile876 from B1b. Further down the pathway, B1b has an arginine (Arg880) at the internal entrance of the pore, whereas A2 and A4 have a glycine (Gly372) and serine (Ser266), respectively, at the equivalent position. The B1b Arg880 side chain extends across the pathway and forms an intracellular gate of CNGA2/A4/B1b. In rod CNGA1/B1, this B1 arginine has been shown to reduce the channel conductance compared to the homomeric CNGA1[17]. It is expected to exert a similar partial blocking effect on the CNGA2/A4/B1b conduction, resulting in a lower channel conductance in the native channel than the CNGA2 homomer[19,21].

## Asymmetrical gating of cAMP-activated CNGA2/A4/B1b

The cAMP-activated open channel structure was determined to reveal the gating mechanism of CNGA2/A4/B1b (Fig. 3a and Supplementary Fig. 6). To eliminate CaM inhibition of the channel, the bound CaM was first removed from the CNGA2/A4/B1b-CaM complex by EGTA treatment, followed by size-exclusion chromatography (Method). cAMP was then added to the purified CNGA2/A4/B1b before EM grid preparation. The cAMP-bound open structure was determined to a lower resolution of 3.59 Å, likely caused by the cAMP-induced dynamic movement at the cytosolic domain (Fig. 3a and Supplementary Fig. 6). The density from the bound cAMP is visible in all CNBDs, which are highly conserved among all subunits (Fig. 3b and Supplementary Fig. 7a). The purine base of the bound-cAMP in CNGA2/A4/B1b adopts an anti conformation similar to cAMP binding observed in CNBDs of rod CNGA1/B and HCN channel (Fig. 3b and Supplementary Fig. 7)[17,37]. It is worth noting that cGMP binds to CNG or HCN channels in a syn conformation (Supplementary Fig. 7b)[17,32,37,38].

Reminiscent to CNGA1/B1, the pore opening of CNGA2/A4/B1b is also asymmetrical upon cAMP activation with the central gating residues from each subunit undergoing distinct conformational changes: Phe364 and Val368 in $A2_L$ undergo large rotation and translation movements away from the central axis; the equivalent gating residues on $A2_R$ barely move; Phe872 and Ile876 in B1b undergo a translation movement without rotation; Phe258 and Met262 in A4 undergo a large rotation with a small translation (Fig. 3c–f)[17]. As discussed below, this asymmetric gating is because the cAMP-induced conformational changes in the cytosolic gating apparatus are translated to different pore-opening S6 movements among all subunits in the native olfactory CNG.

Like cGMP activation in rod CNG, cAMP binding at the CNBDs of olfactory CNGA2/A4/B1b induces a similar cascade of conformational changes that propagate from the cytosolic CNBD of one subunit to the pore-lining S6 helix of the neighboring subunit (Fig. 4a)[17,32]. It starts from an upward swinging of the C-helix closer to the nucleotide-binding pocket, followed by an upward translation of helix C'-turn-helix D' in the C-linker domain. Through the inter-subunit elbow-on-shoulder packing between two neighboring C-linkers, the upward translation of helix C'-turn-helix D' (shoulder) from one subunit propagates to an up and outward movement at helix A'-turn-helix B' (elbow) of the neighboring subunit[37]. In the homomeric CNGA1, the elbow movement is coupled to a homogeneous pore-opening rotation and dilation movement at each S6 through the tight covalent connection between S6 and helix A'[32]. In the heteromeric CNGA2/A4/B1b, however, this coupled movement between helix A'-turn-helix B' elbow and S6 becomes heterogeneous (Fig. 4b-e and Supplementary Movie 1), resulting in a distinct S6 movement in each subunit and consequently asymmetrical pore opening. In the $A2_R$ subunit, the upward and outward movements of its elbow are smaller than those in other subunits, and its S6 barely moves upon cAMP activation. While the elbows in the other three subunits undergo similar movements, the coupled gating conformational changes at their respective S6 helices are quite different: $A2_L$ S6 undergoes both dilation and rotation movements; B1b S6 dilates but does not rotate; A4 S6 rotates but barely dilates.

## CLZ domain assembly and CaM binding

In rod CNGA1/B1, the C-terminal leucine zipper (CLZ) domains of three A1 subunits form an exceptionally long 3-helix coiled coil underneath the CNBDs with a relatively shorter helix from the B1 subunit attached to it, revealing the structural basis of the 3:1 subunit stoichiometry of the native rod CNG channel[13,17]. A similar protein density of the CLZ coiled coil at the equivalent location was also observed in the closed CNGA2/A4/B1b-CaM complex structure. As expected, this CLZ region is poorly defined in the EM-map due to its flexibility. A focused local refinement with a mask around this region was performed to provide a better-resolved map (Fig. 5a,b and Supplementary Fig. 2). Despite the low resolution, the local-refined map reveals the structure features of a long three-helix coiled-coil from the CLZ domains of A2 and A4, a relatively shorter helix from B1b attached to the coiled-coil at about 60° angle, and a bound CaM (Fig. 5b). Among the three coiled-coil helices, a longer one is directly connected to the C-helix of the A4 subunit and has to be the A4 CLZ domain; the other two with equivalent lengths are therefore from the A2 CLZ domains. The B1b helix attached to the coiled-coil was initially designated as the C-terminal helix D (after CNBD helix C) in our earlier study of native rod CNGA1/B1[17]. As discussed in the next section, this helix is more likely from the N-terminal region of the B1b and contains the sequence of the previously defined N-terminal CaM binding site (CaM1 site). It is therefore reassigned as the CaM1 helix in this study (Fig. 5b,c). The lower half of the CaM1 helix directly interacts with CaM, whereas its upper half makes direct contact with the CLZs of $A2_L$ and A4. The structure of a $Ca^{2+}$-bound CaM fits nicely in the CaM density with its C-lobe positioned near the middle of the coiled-coil and its N-lobe positioned near the C-terminal end of the coiled-coil (Fig. 5c). In this docked CaM model, the C-lobe G helix runs parallel to the CLZ helices from $A2_R$ and A4 with close contact between them, and the two C-lobe EF-hands wrap around the lower half of the CaM1 helix. In addition, we observed a stretch of protein density enclosed by the CaM N-lobe. As discussed below, this density is likely from a helix at the C-terminal region of B1b that forms the second CaM binding site (CaM2) and therefore is designated as the CaM2 helix (Fig. 5b,c).

## Structural basis of CaM binding and inhibition

While we can visualize the overall helical assembly of CNGA2/A4/B1b and CaM at the CLZ region in the local-refined map, the molecular identity of the CaM-binding sites cannot be unambiguously defined

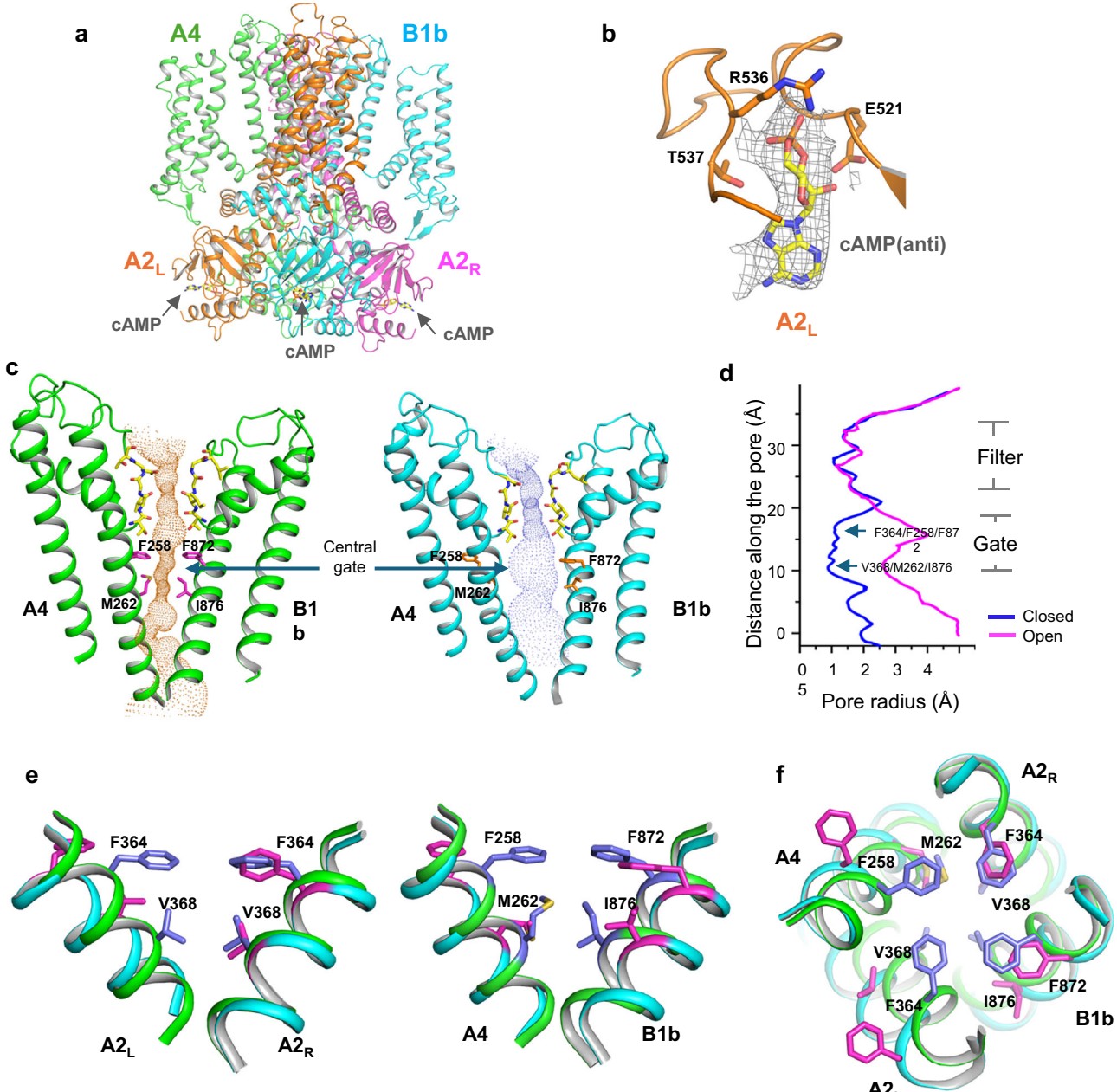

**Fig. 3 | Asymmetrical opening of CNGA2/A4/B1b pore. a** Overall structure of the cAMP-bound open CNGA2/A4/B1b. **b** Zoomed-in view of cAMP binding in A2L subunit with ligand density contoured at 6σ. Same cAMP binding in anti-conformation is observed in all other subunits (Fig. S6A). **c** The ion conduction pore of the apo closed (left) and cAMP-bound open (right) CNGA2/A4/B1b with only A4 and B1b subunits shown for clarity. The central ion pathway is marked with a dotted mesh. Key gating and filter residues are shown as sticks. **d** Pore radius along the central axis in the open and closed CNGA2/A4/B1b. Source data are provided as a Source Data file. **e** Side-view of the structural comparison at the CNGA2/A4/B1b central gate between the open (cyan, with gating residues in magenta) and closed (green, with gating residues in blue) states. Only the S6 helices and gating residues from two diagonal subunits are shown in the superposition. **f** Structural comparison at the central gate between the open (cyan) and closed (green) states viewed from the extracellular side.

due to the low resolution. To this end, we utilized AlphaFold prediction, cross-linking mass spectrometry, and functional analysis of Ca²⁺/CaM inhibition to define the molecular basis of CaM binding in CNGA2/A4/B1b.

Provided with the same protein sequences of CNGA2, A4, and B1b subunits used for structural determination and the 2:1:1 subunit stoichiometry information, AlphaFold predicts an almost identical CaM/CNG assembly at the CLZ region as observed in our structure (Supplementary Fig. 8a, b). Despite incorrect orientation relative to the core part of the channel, the predicted assembly can be directly docked into the map by rigid body movement with minor adjustment,

allowing us to model the assembly based on the prediction (Fig. 5c and Supplementary Fig. 8c). The helix that leans against the CLZ coiled-coil consists of residues 557–587 from the N-terminal region of the B1b subunit rather than the initially suggested C-terminal region in our earlier study[17]; the sequence of this helix overlaps with the previously predicted CaM1 site on B1 subunit and is therefore re-assigned as B1b CaM1-helix (Supplementary Fig. 4)[24,28,30]. The CaM1 helix runs anti-parallel to the coiled-coil with its N-terminal half (residues 557–575) making extensive interaction with the C-lobe of CaM (Fig. 5c and Supplementary Fig. 4). To verify the importance of the CaM1 helix for CaM binding and inhibition, we measured the channel activity of two

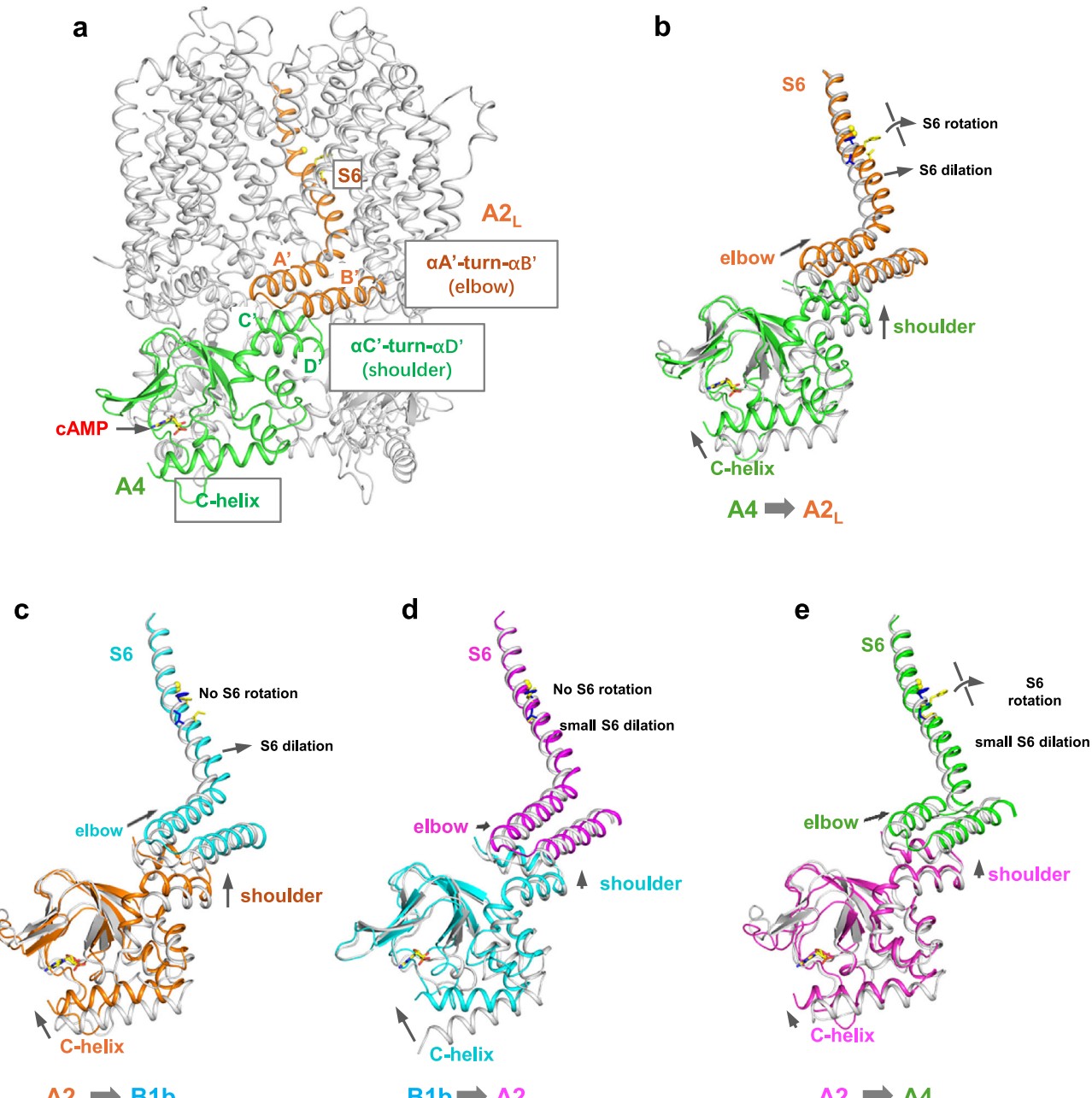

**Fig. 4 | cAMP activation of CNGA2/A4/B1b. a** Structure of cAMP-bound CNGA2/A4/B1b with the key regions important for relaying global gating conformational changes between two neighboring subunits highlighted in color. The key regions from A4 and A2L subunits are highlighted here as an example. CNBD and αC'-turn-αD' from one subunit (A4) are colored in green; αA'-turn-αB' and S6 from the neighboring subunit (A2L) are in cyan. **b** Structural comparison between the closed (grey) and open CNGA2/A4/B1b (colored) illustrating the relay of gating conformational changes from A4 (green) to A2L (orange). Arrows indicate the cAMP-induced movements relayed from the C-helix of A4 to the S6 of A2L. The arrow length marks the amplitude of the movement. **c** The relay of gating conformational changes from A2L (orange) to B1b (cyan). **d** The relay of gating conformational changes from B1b (cyan) to A2R (magenta). **e** The relay of gating conformational changes from A2R (magenta) to A4 (green).

CNGA2/A4/B1b mutants with different N-terminal deletions in B1b: a shorter pre-CaM1 deletion mutant (B1NΔ558) and a longer post-CaM1 deletion mutant (B1NΔ593) (Fig. 6a). Ca²⁺/CaM inhibition is abolished only in the longer deletion mutant with CaM1 helix removed, confirming the essential role of CaM1 helix for CaM binding (Fig. 6b, d).

Corresponding to the stretch of density enclosed by the N-lobe of CaM, AlphaFold predicts a short helix consisting of residues 1107−1119 at the C-terminal region of B1b (Fig. 5c and Supplementary Fig. 4). Engaging in extensive interactions with CaM, this helix likely serves as the second CaM-binding site (CaM2 site) in B1b. Interestingly, an earlier

study suggested a CaM2 site with residues 1129-1143 at a predicted helix further down towards the C-terminus (Supplementary Fig. 4). To verify which of these two is the real CaM2 site, we generate three CNGA2/A4/B1b mutants with C-terminal deletions at the B1b subunit (Fig. 6a): the B1CΔ1087 mutant with both AlphaFold-predicted and the previously suggested CaM2 sites removed; the B1CΔ1120 mutant with the previously suggested CaM2 site removed; and the B1CΔ1150 mutant with both sites intact. Consistent with an earlier functional study, both B1CΔ1120 and B1CΔ1150 mutants can be rapidly inhibited by Ca²⁺/CaM, demonstrating that removing the previously suggested

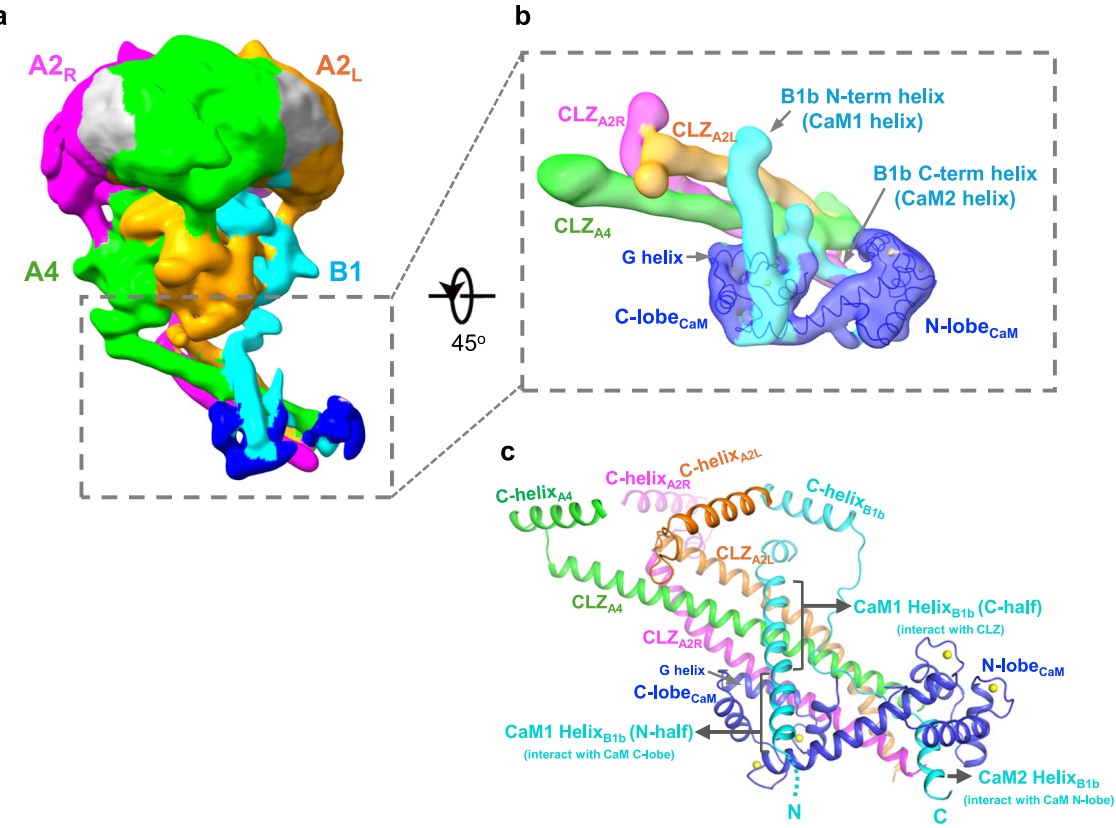

**Fig. 5 | CaM binding at the C-terminal CLZ region of CNGA2/A4/B1b. a** Side view of EM map after focused 3D reconstruction of CNGA2/A4/B1b-CaM complex with the mask around the C-terminal region (boxed region). **b** EM-map at the CLZ region after focused refinement. The structure of Ca²⁺/CaM (blue) is directly docked into the density. **c** Cartoon representation of the structure assembly between CNGA2/A4/B1b and Ca²⁺/CaM at the CLZ region. The modeling of the assembly is facilitated by the AlphaFold prediction.

CaM2 site does not affect CaM inhibition[24]. Only the longest deletion (B1CΔ1087 mutant) with the AlphaFold-predicted CaM2 site removed abolishes the CaM inhibition, confirming that the predicted short helix from residues 1107-1119 is the bona fide CaM2 site (Fig. 6c, d). To ensure that the loss of CaM inhibition in B1NΔ593 and B1CΔ1087 is not caused by incorrect subunit assembly of the mutant channel, we also measured the cAMP activation and Ca²⁺ blockage properties of these two mutants, which are similar to the wild-type CNGA2/A4/B1b, confirming the correct 2:1:1 subunit assembly of the mutants (Supplementary Fig. 9).

In addition to using AlphaFold to facilitate our model building, we also performed cross-linking mass spectrometry using the amine-to-amine bis(sulfosuccinimidyl)suberate (BS3) cross-linker to investigate the protein-protein interactions between CNGA2/A4/B1b and CaM (Methods). Two lysine residues on CaM (K22 in the N-lobe and K95 in the C-lobe) participate in the inter-subunit cross-linking reactions, and they are exclusively cross-linked to the B1b subunit, confirming the close association between CaM and B1b (Fig. 6e and Supplementary Fig. 8d). Both K22 and K95 can be cross-linked to multiple C-terminal lysine residues on the B1b subunit after the CaM2 helix, indicating that the post-CaM2 region has the flexibility to reach both lysine residues in the N- and C-lobe. Among the unambiguously identified inter-subunit cross-linking reactions, only one lysine from the N-terminal region of B1b interacts with CaM. That is between K514 before the B1b CaM1 helix and K95 in the C-lobe of CaM, consistent with a close contact between the C-lobe and the CaM1 helix in our structural model. In addition, the cross-linking reaction between K1118 at the C-terminal end of the CaM2 helix and K22 confirms the close contact between the B1b CaM2 helix and the N-lobe of CaM.

## Discussion

We present the structures of the olfactory CNG channel in its cAMP-bound open state and CaM-bound closed state, providing structural insights into cAMP-induced activation and CNGB1b-mediated CaM binding of the channel. In its native assembly, the olfactory CNGA2/A4/B1b heterotetrameric complex features two diagonally positioned A2 subunits with the modulatory A4 and B1b subunits occupying the other two diagonal positions. Upon cAMP activation, each subunit undergoes distinct conformational changes, resulting in asymmetrical gate opening reminiscent of the native rod CNGA1/B1 channel[17].

Combining the structural and functional analyses with AlphaFold prediction, we defined the CaM binding at the CLZ coiled-coil region and elucidated the long-sought structural mechanism underlying the Ca²⁺/CaM inhibition of the native olfactory CNG, a physiologically important feedback process for odor adaptation. Upon Ca²⁺ binding, CaM modulates the native olfactory CNG activity by mainly interacting with the B1b subunit, with its C-lobe binding to the B1b N-terminal CaM1 site and its N-lobe binding to the B1b C-terminal CaM2 site. Deleting either site on B1b can abolish the CaM inhibition of the olfactory CNG, as demonstrated in our functional assay. In addition, the B1b CaM1 helix is also tightly packed against the 3-helix coiled-coil formed by the CLZ domains from the A2 and A4 subunits. Consequently, Ca²⁺/CaM binding yields a sub-domain assembly, consisting of CaM, CaM1, and CaM2 helices from B1b, and the CLZ 3-helix coiled-coil from A2 and A4, right underneath the C-helices of CNBDs. As the channel activation is initiated by the upward movements of C-helices that are directly connected to the CLZ domains with short loops, this stable sub-domain assembly and its tight connection to the C-helices would restrain their movements and inhibit the cAMP activation of the

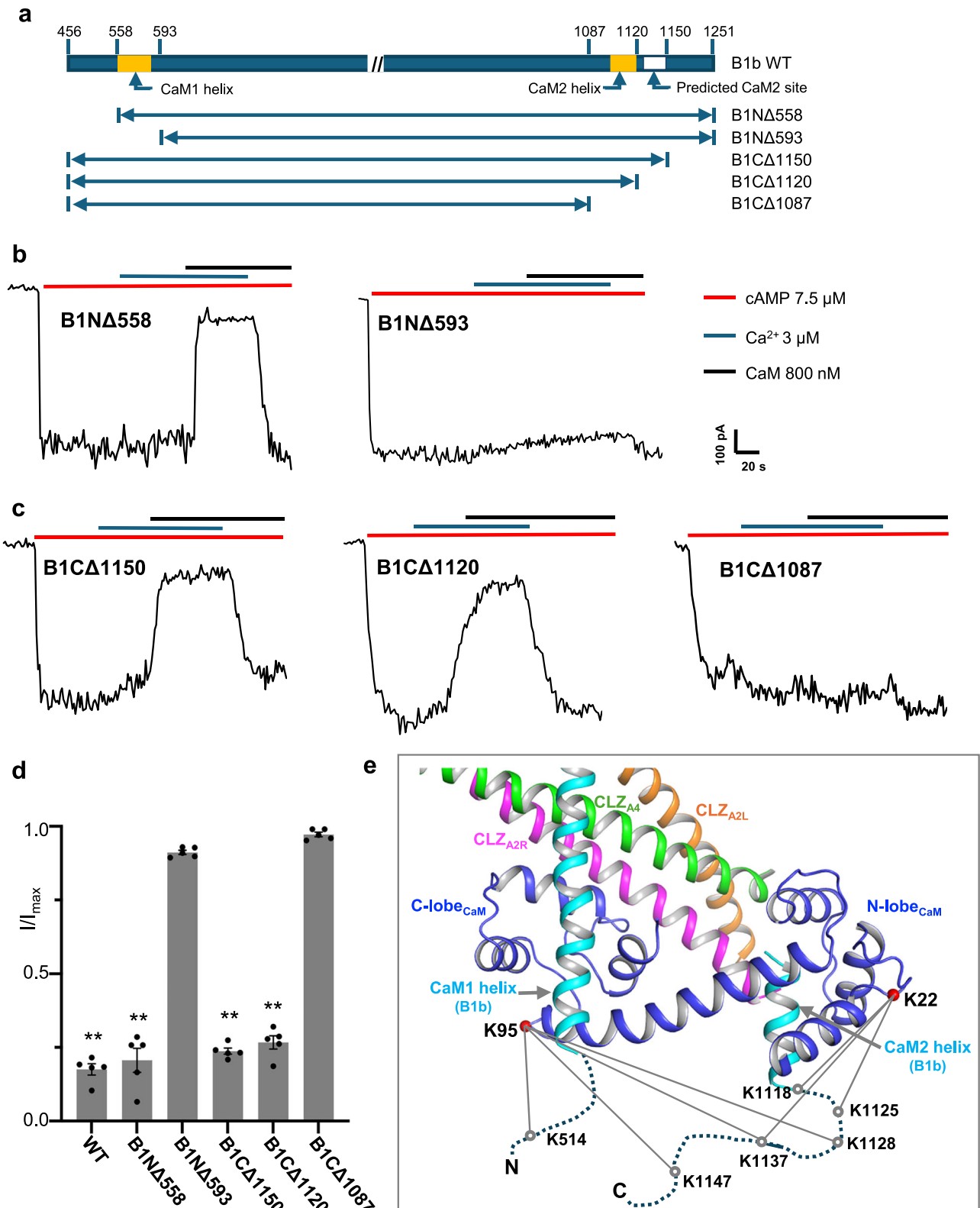

**Fig. 6 | N- and C-terminal CaM-binding sites on CNGB1b. a** Design of N- and C-terminal deletion constructs of CNGB1b for the Ca²⁺/CaM inhibition test. **b** Sample traces of Ca²⁺/CaM inhibition of olfactory CNG with B1b N-terminal deletion mutations recorded in excised patches. Currents were recorded after endogenous CaM washout. Inward current at −100mV was elicited by 7.5 μM cAMP. **c** Sample traces of Ca²⁺/CaM inhibition of olfactory CNG with B1b C-terminal deletion mutations recorded in excised patches. **d** Statistics of Ca²⁺/CaM inhibition of CNGA2/A4/B1b WT and mutant channels. Currents are normalized against the initial cAMP-activated inward currents at −100 mV before adding Ca²⁺/CaM. Data points are mean ± SEM (*n* = 5 independent replicates, *p*-values were calculated using a two-sided Student's *t*-test, **\**p* < 0.01). Source data are provided as a Source Data file. **e**, Inter-subunit cross-linking reactions between CaM and CNGB1b.

channel. One unique feature of native olfactory CNG is that CaM remains tightly bound to the channel even at low $Ca^{2+}$ concentration, likely through one of the CaM sites on B1b. This pre-association between CaM and the native olfactory CNG would allow CaM to rapidly engage in channel inhibition upon elevating the $Ca^{2+}$ level[24].

Interestingly, previous studies demonstrated that CaM could also bind to the CNGA2 homomer and inhibit its channel activity, but with much slower kinetics than the CaM inhibition of the native olfactory CNG[21,25]. A CaM binding site was identified at the N-terminal region of the A2 subunit, whose deletion can abolish CaM inhibition of the CNGA2 homomer[31,39]. However, the equivalent deletion of the CaM site on the A2 subunit does not affect CaM inhibition of the native olfactory CNG channel[24]. This is consistent with our structural analysis showing that CaM binding in the native channel does not involve the A2 subunit. How CaM binds to the homomeric CNGA2 channel remains an open question and warrants further study. It is possible that in the absence of B1b, the N-terminal CaM site from one A2 subunit can interact with the C-terminal CLZ 3-helix coiled-coil from the other three A2 subunits and substitute the B1 CaM1 helix for CaM binding.

## Methods

### Protein expression and purification

A tri-cistronic construct containing the human genes CNGA2, CNGA4, and CNGB1b, with each pair of neighboring genes linked by a P2A self-cleaving peptide, was used for the expression of heterotetrameric channel proteins[34]. CNGB1b in the construct is a shorter variant of CNGB1, comprising amino acids 456-1251. A Flag tag was placed at the N-terminus of all three subunits, and the final tri-cistronic construct was cloned into the pEZT-BM vector[40]. The human gene CaM, with a C-terminal Strep tag, was cloned into the pEZT-BM vector. Proteins were expressed in HEK293F cells using the BacMam system (Thermo Fisher Scientific). Bacmids were synthesized using *E. coli* DH10bac cells (Thermo Fisher Scientific), and baculoviruses were produced in Sf9 cells using Cellfectin II reagent (Thermo Fisher Scientific). For protein expression, cultured HEK293F cells were infected with baculoviruses at a ratio of 1: 4: 80 (virus CaM: virus CNG: HEK293F, v/v) for 10 h. 10 mM sodium butyrate was then introduced to boost protein expression, and the cells were cultured in suspension at 30 °C for another 60 h and harvested by centrifugation at 4,000xg for 15 min. All purification procedures were carried out at 4 °C. The cell pellet was resuspended in buffer A (25 mM HEPES pH 7.4, 200 mM NaCl, 2 mM $CaCl_2$) supplemented with a protease inhibitor cocktail (2 µg/ml DNase, 0.5 µg/ml pepstatin, 2 µg/ml leupeptin, 1 µg/ml aprotinin, and 0.1 mM PMSF). Following homogenization by sonication, CNGA2/A4/B1b was extracted with 1% (w/v) Lauryl Maltose Neopentyl Glycol (LMNG, Anatrace) by gentle agitation for 2 h. The supernatant was collected by centrifugation at 40,000xg for 30 min and incubated with anti-Flag M2 affinity resin (Genescript) by gentle agitation for 1 h. The resin was then transferred to a disposable gravity column (Bio-Rad) and washed with 20 column volumes of buffer A supplemented with 0.03% (w/v) LMNG, followed by 20 column volumes of buffer A supplemented with 0.06% (w/v) digitonin. The protein was eluted in buffer A containing 0.06% (w/v) digitonin and 0.2 mg/ml Flag peptide. The eluate was subsequently incubated with Strep-Tactin affinity resin (IBA) for 1 h. The resin was washed with 30 column volumes of buffer A supplemented with 0.06% (w/v) digitonin, and the protein was eluted in wash buffer containing 50 mM biotin. The CNG-CaM protein eluate was concentrated and further purified by size-exclusion chromatography on a Superose 6 10/300 GL column (GE Healthcare) in buffer A with 0.06% (w/v) digitonin. Peak fractions were collected and concentrated to ~ 3.5 mg/ml for cryo-EM analysis. To remove bound CaM, the purified CNG-CaM complex was incubated with 2 mM EGTA for 1 h, followed by further purification by size-exclusion chromatography on a Superose 6 10/300 GL column (GE Healthcare) in buffer B (25 mM HEPES pH 7.4, 200 mM NaCl) supplemented with 0.06% (w/v) digitonin. Peak

fractions were concentrated to ~ 3.5 mg/ml and incubated with 5 mM cAMP for 2 h to prepare the CNG-cAMP complex before being applied to the grids and plunge-frozen.

Mutant constructs for electrophysiological recordings were generated using standard procedures and verified by sequencing.

HEK293 cells (CRL-1573) were purchased from and authenticated by the American Type Culture Collection (ATCC, Manassas, VA). HEK293F cells (RRID: CVCL_D603) were purchased from and authenticated by Thermo Fisher Scientific. All cell lines tested negative for mycoplasma contamination.

### Cryo-EM sample preparation and data acquisition

Purified CNG-CaM or CNG-cAMP samples were applied to glow-discharged Quantifoil R1.2/1.3 300-mesh gold holey carbon grids (Quantifoil, Germany), blotted for 4 s under 100% humidity at 4 °C, and plunge-frozen in liquid ethane using a Vitrobot Mark IV (FEI). Cryo-EM data were collected on a Titan Krios microscope (Thermo Fisher Scientific) operated at 300 kV, equipped with either a K3 Summit direct electron detector and a GIF-Quantum energy filter or a Falcon 4i direct electron detector and a Selectris energy filter, using a 10 eV slit width. Images were recorded using SerialEM at a calibrated pixel size of 0.83 Å or 0.738 Å, with a nominal defocus range of −0.8 to −1.8 µm. Each movie was dose-fractionated into 60 frames at a dose rate of 1 e⁻/Å²/frame, resulting in a total dose of about 60 e⁻/Å².

### Image processing

Cryo-EM data were processed following a general workflow described below with dataset-specific modifications. Movie frames were motion-corrected and dose-weighted using MotionCor2[41], and the CTF parameters were estimated using the GCTF program[42]. Micrographs were then manually inspected to discard images with poor defocus values or visible ice contamination. Particles were picked using crYOLO[43] and extracted with a binning factor of 3 in RELION[44,45]. Extracted particles were subjected to 2D classification, and particles from well-defined classes were selected for the following 3D classification. A low-pass filtered map of the hetero-tetrameric CNGA1/B1 channel EMD-24458 was used as the initial reference for 3D classification[17]. Particles from the best-resolving 3D class were re-extracted at the original pixel size and subjected to heterogenous 3D refinement, non-uniform refinement, CTF refinement, and local refinement. Resolutions were reported based on the gold-standard Fourier shell correlation using the 0.143 criterion[46]. Local resolution estimation was performed using cryoSPARC[47].

For the CNG-CaM dataset, to improve the resolution of the CLZ region, a soft mask was applied around the CLZ region for focused refinement. Subsequently, particle subtraction targeting the CLZ region, followed by 3D refinement,t was performed to further enhance the map quality at the CLZ region.

For the CNG-cAMP dataset, the quality of the EM density maps corresponding to the TM and CNBD domains was further improved through focused refinement, allowing for accurate model building for a major part of the protein.

### Model building, refinement, and validation

Both cryo-EM maps of the CNG-CaM and CNG-cAMP complexes showed high-quality density to allow de novo model building in Coot[48], facilitated by the previous cryo-EM structures of CNGA1/B1 (PDB: 7RH9 [10.2210/pdb7rh9/pdb], 7RHI [10.2210/pdb7rhi/pdb])[17]. Models were manually adjusted in Coot and refined against the summed maps using the phenix.real_space_refine, with secondary structure restraints applied[49]. The model was validated using previously described methods to avoid overfitting[50]. The final models include residues 130-583 of CNGA2, residues 25-477 of CNGA4, and residues 645-748 and 757-1089 of CNGB1. The CLZ-CaM region was modeled based on the predicted AlphaFold structure. The geometric statistics for the refined models

were generated using MolProbity[51]. All structural figures were prepared using PyMol (Schrödinger, LLC.) and UCSF Chimera[52]. Pore radii were calculated using the HOLE program[53].

## Electrophysiology

1.5 µg of the pEZT-BM vector containing wild-type or mutant HsCNGA2/A4/B1b was transfected into HEK293 cells using Lipofectamine 2000 (Life Technologies). To facilitate identification of transfected cells for patch clamp, 0.2 µg of the pNGFP-EU vector containing GFP was co-transfected[54]. After 48 h, cells were dissociated by trypsin treatment, resuspended in complete serum-containing medium, and replated onto 35 mm tissue culture dishes for electrophysiological recording.

Channel currents were recorded using whole-cell and inside-out patch-clamp configurations. Patch pipettes were pulled from borosilicate glass (Harvard Apparatus) and heat-polished to a resistance of 3–5 MΩ. Giga-seals (>10GΩ) were formed by gentle suction, followed by brief suction or zap to establish the whole-cell configuration. For inside-out recording, the pipette was withdrawn from the cell, occasionally exposing the tip to air. Recordings were made using an Axo-Patch 200B amplifier (Molecular Devices) with 1 kHz low-pass analog filter. Data was sampled at a rate of 20 kHz using a Digidata 1550B digitizer (Molecular Devices) and analyzed with pClamp11 software (Molecular Devices). To generate current-voltage (I-V) relationships, the membrane potential was held at 0 mV, followed by voltage pulses ramp from −100 to +100 mV over 800 ms.

To study cAMP activation and CaM inhibition, recordings were performed in an inside-out configuration. The pipette solution contained 140 mM NaCl, 4 mM KCl, 1 mM EGTA, and 10 mM HEPES pH 7.4, whereas the bath solution contained 140 mM KCl, 5 mM EGTA, and 10 mM HEPES pH 7.4. For cAMP activation assays, various concentrations of cAMP were added to the bath. For CaM inhibition, 7.5 µM cAMP was added for channel activation. To assess endogenous CaM-mediated inhibition, the bath solution was supplemented with 3 µM free $Ca^{2+}$, calculated by mixing 1 mM EGTA and 0.98 mM $CaCl_2$ using MAXCHELATOR. For exogenous CaM inhibition, patches were extensively washed with bath solution containing 10 mM EGTA for 10 min to remove endogenous CaM, followed by application of 3 µM $Ca^{2+}$ and 800 nM purified CaM.

Extracellular $Ca^{2+}$ blockage was examined in the whole-cell configuration. The pipette solution contained 140 mM KCl, 5 mM EGTA, 10 mM HEPES, pH 7.4, and 1 mM cAMP. The bath solution contained 140 mM NaCl, 4 mM KCl, 1 mM EGTA, and 10 mM HEPES, pH 7.4. Free $Ca^{2+}$ concentrations in the bath solution were adjusted by mixing 1 mM EGTA with appropriate amounts of $CaCl_2$ calculated using MAXCHELATOR.

All data are presented as mean ± SEM, with the number of independent measurements (n) indicated in the figure legends. Statistical analyses were performed using OriginPro 8 (OriginLab).

## XL-MS chemical cross-linking

The purified CNG-CaM complex at a concentration of 1.5 mg/mL was cross-linked with 1 mM bis (sulfosuccinimidyl) suberate (BS3, Thermo Scientific Pierce) at 4 °C for 1 h. The reaction was quenched by adding 20 mM Tris-HCl (pH 7.5) at room temperature for 10 min. Following the cross-linking reaction, the protein sample was re-purified by size-exclusion chromatography using a Superose 6 10/300 GL column and subsequently subjected to in-gel digestion for mass spectrometry (MS) analysis. Cross-linking analysis was performed using Mascot (v 2.7.0). MS/MS spectra were searched against the SwissProt protein database, restricted to the *Homo sapiens* taxonomy. Trypsin was specified as the proteolytic enzyme, allowing up to 2 missed cleavages. The precursor and fragment ion tolerances were set to 20 ppm and 0.5 Da, respectively. Carbamidomethylation of cysteine was set as a fixed modification, while oxidation of methionine was included as a variable modification.

## Reporting summary

Further information on research design is available in the Nature Portfolio Reporting Summary linked to this article.

## Data availability

The cryo-EM density maps of the human CNGA2/A4/B1b-CaM and CNGA2/A4/B1b-cAMP complex in this study have been deposited in the Electron Microscopy Data Bank under accession numbers EMD-64395 (CNGA2/A4/B1b-CaM) and EMD-64394 (CNGA2/A4/B1b-cAMP), respectively. Atomic coordinates have been deposited in the Protein Data Bank under accession numbers 9UPG [10.2210/pdb9UPG/pdb] (CNGA2/A4/B1b-CaM) and 9UPF [10.2210/pdb9UPF/pdb] (CNGA2/A4/B1b-cAMP). The cross-linking mass spectrometry data have been deposited to the ProteomeXchange Consortium via the MassIVE repository with the dataset identifier PXD063119. Source data are provided with this paper.

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

## Acknowledgements

Single particle cryo-EM data were collected at the University of Texas Southwestern Medical Center Cryo-EM Facility, which is funded by the CPRIT Core Facility Support Award RP170644. Cryo-EM sample grids were prepared at the Structural Biology Laboratory at UT Southwestern Medical Center, partially supported by grant RP170644 from CPRIT. This work was supported in part by the Howard Hughes Medical Institute (to Y.J.) and by grants from the National Institute of Health (R35GM140892 to Y.J.).

## Author contributions

J.X. and N.G. prepared the samples; J.X. performed data acquisition, image processing, and structure determination; W.Z. performed electrophysiology recording; Y.J. supervised the study; all authors participated in research design, data analysis, discussion, and manuscript preparation.

## Competing interests

The authors declare no competing interests.
