## [Transparent Peer Review file · Nature Communications]

Structural mechanisms of assembly, gating, and calmodulin modulation of human olfactory CNG channel

Corresponding Author: Professor Youxing Jiang

Version 0:

Reviewer comments:

Reviewer #1

(Remarks to the Author)

Olfactory CNG channels produce the primary electrical event in olfactory receptors in response to odorant. They are activated by the direct binding of cAMP and modulated by Ca-calmodulin, a process responsible for olfactory adaptation. These complex proteins comprise four main subunits of three different types (CNGA2, CNGA4, and CNGB1b) in a 2:1:1 stoichiometry. This manuscript presents the high resolution cryo-EM structures of these proteins in a calmodulin-inhibited closed state and a cAMP activated open state. The results are very illuminating. They reveal an amazing complexity in the cAMP-induced opening conformational change, where the conformational changes in each subunit are different (even the two CNGA2 subunits with identical sequence). In addition, they reveal the structural basis for the assembly of the heterotetramers, and for the Ca-calmodulin inhibition of the channel, which involves previously unknown structural elements and bridging of two disparate parts of the CNGB1b subunit. These structures provide long sought after answers and reveal many unimagined complexities.

The paper is extremely well written, and all of the results are clearly presented and convincing. I am particularly amazed how the pore segments in the heterotetramer exhibit different conformational changes, largely determined by the identity of their neighboring subunit through an intersubunit "elbow-on-shoulder" interaction. While these authors have shown a similar result in the native rod CNG channel, it is more complex in the olfactory channel with the CNGA4 subunit. In addition, the findings that the CNGA4 CLZ domain contributes to the three helix bundle, and the CAM1 domain on the N-terminus of CNGB1b interacts with this bundle are novel and interesting. Finally, it is fascinating that Ca-calmodulin inhibits the channel by bridging the N- and C-terminal CAM binding domains in the CNGB1b subunit. In rod CNG channels, the inhibition (presumably also from the CNGB1 subunit), also required extensive washing with EDTA to remove the endogenous Ca regulatory protein (Gordon et al., 1995). Does the subsequent addition of Ca-calmodulin also require extensive washing with EDTA to reverse? It might be worth commenting on as it might suggest that the native regulator is a calmodulin-like protein but not calmodulin or is bound with other auxiliary proteins that also wash off. Overall, these experiments answer many questions and provide the framework for many future experiments.

Reviewer #2

(Remarks to the Author)

The manuscript by Xue et al is a substantial contribution to our understanding of sensory transduction processes including structural underpinnings of the novel evolutionary adaptations and genetic selections that afford visual and olfactory systems their properties and challenges. The authors utilize electrophysiology and mutational analysis to confirm specific aspects of their structural and modeling studies. The interpretations are carefully drawn and appropriate weaknesses and unanswered questions addressed.

There is one physiological aspect of the novel subunit composition of the native olfactory that the authors appear to largely or completely ignore. Specifically, it was recognized in the first characterizations of CNGA2 that the EC50 for activation of the native channel by cAMP (the physiologic ligand) is ~10 lower than the homomeric CNGA2 channel while the EC50 for cGMP is low and nearly identical in the two multimeric forms. In contrast, expression of the visual CNGA1 homomultimer in comparison to the native multimer revealed no such variation. Given the nicely and clearly described unique interactions of each subunit's CNBD on the adjacent subunit to control gating, it would seem the authors could address this question with their structural insights or at least speculation.

Minor points

Consider either coloring or labeling the additional visible cAMP molecules in figure 3a

Reviewer #3

(Remarks to the Author)

In this excellent work, the authors present two structures of the human olfactory CNG channel, the apo closed channel bound with CaM and the open channel bound with cAMP. The structures reveal how the olfactory CNG channel is assembled and confirm the 2:1:1 subunit stoichiometry concluded in previous biochemical and functional studies; they also provide new mechanistic insights into ion permeation, gating and regulation by CaM of the olfactory CNG channel. The structures are overall solid and convincing. They are the first structures of this subclass of CNG channels and therefore represent a significant breakthrough in the ion channel field in general and the CNG channel field in particular. The structures mark a new starting point for further structural and functional studies on these physiologically important channels.

I have only one concern. The authors state in Discussion that their structures reveal "the structural mechanisms of cAMP activation and CaM inhibition" (line 314, 315). I think this statement is not quite fully supported by the two structures. The CaM-bound structure shows only where CaM binds but does not show HOW CaM inhibits the channel. The authors speculated on how this might happen, but it is just that - speculation. To better understand how CaM works, one would like to see the structure of the apo channel without CaM and the structure of the cAMP-bound channel with CaM. Did the authors try to get these structures? If not, it might be worthwhile to try. If yes, what were the results? Without these two additional structures, the authors need to tone down the conclusions.

Version 1:

Reviewer comments:

Reviewer #1

(Remarks to the Author)

The authors have more than adequately addressed my comments.

Reviewer #2

(Remarks to the Author)

Thank you for clarifying the issues that prevent additional insights into CAMP affinity.

Additionally, the location of the cMAP molecules in the structure presented is now much easier for the reader to appreciate.

Reviewer #3

(Remarks to the Author)

The authors have satisfactorily addressed my concern.

To Reviewer #1

Olfactory CNG channels produce the primary electrical event in olfactory receptors in response to odorant. They are activated by the direct binding of cAMP and modulated by Ca-calmodulin, a process responsible for olfactory adaptation. These complex proteins comprise four main subunits of three different types (CNGA2, CNGA4, and CNGB1b) in a 2:1:1 stoichiometry. This manuscript presents the high resolution cryo-EM structures of these proteins in a calmodulin-inhibited closed state and a cAMP activated open state. The results are very illuminating. They reveal an amazing complexity in the cAMP-induced opening conformational change, where the conformational changes in each subunit are different (even the two CNGA2 subunits with identical sequence). In addition, they reveal the structural basis for the assembly of the heterotetramers, and for the Ca-calmodulin inhibition of the channel, which involves previously unknown structural elements and bridging of two disparate parts of the CNGB1b subunit. These structures provide long sought after answers and reveal many unimagined complexities.

The paper is extremely well written, and all of the results are clearly presented and convincing. I am particularly amazed how the pore segments in the heterotetramer exhibit different conformational changes, largely determined by the identity of their neighboring subunit through an intersubunit “elbow-on-shoulder” interaction. While these authors have shown a similar result in the native rod CNG channel, it is more complex in the olfactory channel with the CNGA4 subunit. In addition, the findings that the CNGA4 CLZ domain contributes to the three helix bundle, and the CAM1 domain on the N-terminus of CNGB1b interacts with this bundle are novel and interesting. Finally, it is fascinating that Ca-calmodulin inhibits the channel by bridging the N- and C-terminal CAM binding domains in the CNGB1b subunit. In rod CNG channels, the inhibition (presumably also from the CNGB1 subunit), also required extensive washing with EDTA to remove the endogenous Ca regulatory protein (Gordon et al., 1995). Does the subsequent addition of Ca-calmodulin also require extensive washing with EDTA to reverse? It might be worth commenting on as it might suggest that the native regulator is a calmodulin-like protein but not calmodulin or is bound with other auxiliary proteins that also wash off. Overall, these experiments answer many questions and provide the framework for many future experiments.

We greatly appreciate Reviewer #1's positive feedback. In response to the suggestion, we performed the proposed experiment. The result shows that calmodulin added after the initial washout also binds tightly to the olfactory CNG channel and requires an extended period of washout with EGTA-containing solution to be removed (note that we used EGTA instead of EDTA in all our experiments). We have included this result in the revised manuscript. Furthermore, the initial Ca²⁺-dependent inhibition is abolished when either the CaM1 or CaM2 site on CNGB1b is deleted. These findings strongly support the conclusion that the initial inhibition of the olfactory CNG channel is mediated by endogenous calmodulin - a mechanism that appears to differ from that reported for rod CNG channels in Gordon's 1995 study.

To Reviewer #2

The manuscript by Xue et al is a substantial contribution to our understanding of sensory transduction processes including structural underpinnings of the novel evolutionary adaptations and genetic selections that afford visual and olfactory systems their properties and challenges. The authors utilize electrophysiology and mutational analysis to confirm specific aspects of their structural and modeling studies. The interpretations are carefully drawn and appropriate weaknesses and unanswered questions addressed.

There is one physiological aspect of the novel subunit composition of the native olfactory that the authors appear to largely or completely ignore. Specifically, it was recognized in the first characterizations of CNGA2 that the EC50 for activation of the native channel by cAMP (the physiologic ligand) is ~10 lower than the homomeric CNGA2 channel while the EC50 for cGMP is low and nearly identical in the two multimeric forms. In contrast, expression of the visual CNGA1 homomultimer in comparison to the native multimer revealed no such variation. Given the nicely and clearly described unique interactions of each subunit's CNBD on the adjacent subunit to control gating, it would seem the authors could address this question with their structural insights or at least speculation.

We appreciate Reviewer #2's positive feedback and constructive suggestions. We are aware of the reported variation in cAMP EC₅₀ between homomeric CNGA2 and native olfactory CNG channels, and such variation is not observed for cGMP. We briefly mentioned the ~10-fold difference in cAMP EC₅₀ between the CNGA2 homomer and the native olfactory CNG in the introduction but did not elaborate further. We were cautious not to speculate on the structural basis for the differences in ligand efficacy among different CNG forms for the following reasons: the CNBD appears more dynamic upon cAMP binding, resulting in lower local resolution in this region; while we can detect overall cAMP-induced conformational changes at the CNBDs, the molecular details of inter-subunit contacts between neighboring CNBDs remain poorly defined compared to other regions of the channel; furthermore, the absence of a high-resolution structure of homomeric CNGA2 limits our ability to compare inter-subunit contacts between the homotetrameric CNGA2 and the heterotetrameric CNGA2/A4/B1b. As a result, our current data do not allow us to determine whether the observed differences in cAMP EC₅₀ arise from variations in binding affinity among subunits or from differences in the coupling between the CNBD and the pore in the heteromeric channel.

Minor points:

Consider either coloring or labeling the additional visible cAMP molecules in figure 3a.

As suggested, the additional visible cAMP molecules are labeled.

To Reviewer #3

In this excellent work, the authors present two structures of the human olfactory CNG channel, the apo closed channel bound with CaM and the open channel bound with cAMP. The structures reveal how the olfactory CNG channel is assembled and confirm the 2:1:1 subunit stoichiometry concluded in previous biochemical and functional studies; they also provide new mechanistic insights into ion permeation, gating and regulation by CaM of the olfactory CNG channel. The structures are overall solid and convincing. They are the first structures of this subclass of CNG channels and therefore represent a significant breakthrough in the ion channel field in general and the CNG channel field in particular. The structures mark a new starting point for further structural and functional studies on these physiologically important channels.

I have only one concern. The authors state in Discussion that their structures reveal "the structural mechanisms of cAMP activation and CaM inhibition" (line 314, 315). I think this statement is not quite fully supported by the two structures. The CaM-bound structure shows only where CaM binds but does not show HOW CaM inhibits the channel. The authors speculated on how this might happen, but it is just that - speculation. To better understand how CaM works, one would like to see the structure of the apo channel without CaM and the structure of the cAMP-bound channel with CaM. Did the authors try to get these structures? If not, it might be worthwhile to try. If yes, what were the results? Without these two additional structures, the authors need to tone down the conclusions.

We thank Reviewer #3 for the positive feedback on our manuscript. We agree with the reviewer's constructive suggestion that additional structural data are necessary to fully elucidate the mechanisms of cAMP activation and CaM-mediated inhibition. During this study, we also determined the apo structure of the channel without bound CaM, using a construct with an N-terminal deletion in the A2 subunit. Although this structure was resolved at lower resolution, the channel region is virtually identical to that of the full-length channel in complex with CaM. For this reason, we included only the full-length structure in the manuscript. We have not yet attempted to determine a cAMP-bound structure of the olfactory CNG channel in complex with CaM, but we agree that this is a valuable direction for future investigation. In line with the reviewer's suggestion, we have moderated the conclusion statement in the revised **Discussion** section.

REVIEWERS' COMMENTS

Reviewer #1 (Remarks to the Author):

The authors have more than adequately addressed my comments.

We appreciate the support from this reviewer.

Reviewer #2 (Remarks to the Author):

Thank you for clarifying the issues that prevent additional insights into cAMP affinity. Additionally, the location of the cAMP molecules in the structure presented is now much easier for the reader to appreciate.

We appreciate the reviewer's supportive comments.

Reviewer #3 (Remarks to the Author):

The authors have satisfactorily addressed my concern.

We appreciate the support from this reviewer.